# Various Stages of Immune Synapse Formation Are Differently Dependent on the Strength of the TCR Stimulus

**DOI:** 10.3390/ijms21072475

**Published:** 2020-04-02

**Authors:** Michael Estl, Pascal Blatt, Xuemei Li, Ute Becherer, Hsin-Fang Chang, Jens Rettig, Varsha Pattu

**Affiliations:** Cellular Neurophysiology, Center for Integrative Physiology and Molecular Medicine (CIPMM), Saarland University, 66421 Homburg, Germany; estlmichi@gmx.at (M.E.); pascalblatt@gmx.net (P.B.); S8xxliii@stud.uni-saarland.de (X.L.); ute.becherer@uks.eu (U.B.); hsin-fang.chang@uks.eu (H.-F.C.); jrettig@uks.eu (J.R.)

**Keywords:** cytotoxic T lymphocyte(s), immune synapse, supramolecular activation center, cytotoxic granule(s), lipid bilayers, total internal reflection fluorescence microscopy

## Abstract

Cytotoxic T lymphocytes (CTL) are key players of the adaptive immune system that target tumors and infected cells. A central step to that is the formation of a cell–cell contact zone between the CTL and its target called an immune synapse (IS). Here, we investigate the influence of the initial T cell receptor (TCR) trigger of a cytolytic IS on the distinct steps leading to cytotoxic granule (CG) exocytosis. We stimulated primary CTLs from mouse using lipid bilayers with varying anti-CD3 but constant ICAM concentrations. We fluorescently labeled molecular markers of distinct IS zones such as actin, CD3, granzyme B, and Synaptobrevin2 in CTLs and imaged cytolytic IS formation by total internal reflection fluorescence microscopy (TIRFM). We found that an intermediate anti-CD3 concentration of 10 µg/mL induces the fastest adhesion of CTLs to the bilayers and results in maximal CG fusion efficiency. The latency of actin ring formation, dwell time, and maximum surface area at the IS exhibit different dependencies on the stimulatory anti-CD3 concentrations. The number and surface area of CD3 clusters at the IS seem to show a different dependency to the TCR trigger when compared to their dwell time. Finally, the mode of full CG exocytosis appears to be independent of the TCR trigger.

## 1. Introduction

Cytotoxic T lymphocytes (CTLs) are part of the adaptive immune system and function to kill infected and tumorigenic cells. Similar to helper T cells, they first need to be activated via the T cell receptor which forms a complex with antigenic major histocompatibility complexes (MHC) on the target cells. This triggers a signaling cascade inside the T cell leading to the formation of an immune synapse (IS) [1]. Proximal TCR signaling is mainly led by two kinases Lck and Fyn. The downstream cascade of Lck activation encompassing the proximal TCR signaling includes ZAP70, LAT, SLP76, phospholipase C (PLCγ1). ZAP70 phosphorylates LAT which recruits SLP76 which in turn is phosphorylated by ZAP70 to result in the LAT-SLP76 signalosome. One of the most important signaling effectors of the LAT-SLP 76 signalosome is phospholipase C (PLC) which transduces the TCR signal by hydrolyzing phosphatidyl (4,5) bisphosphate into IP3 and DAG. The local DAG gradient promotes recruitment of PKC isoforms which mediate signaling inducing MTOC polarization and synapse polarity [2]. IP3 binding to IP3 receptors on the membrane of the endoplasmic reticulum initiates Ca^2+^ depletion from the ER lumen which triggers Ca^2+^ influx from the extracellular space. This store operated calcium entry (SOCE) is essential for CTL effector function [3].

IS formation results in visible morphological modifications occurring due to changes in the composition and density of molecules such as central actin depletion, recycling TCR polarization, cytotoxic granule (CG) polarization [4]. Initial triggering of the TCR leads to formation of numerous TCR micro-clusters whose structure resembles a miniature synapse in that every TCR micro-cluster is surrounded by a micro-ring containing adhesion molecules. These micro-adhesion ring TCR clusters are important sites for early TCR signaling and coalesce to form larger clusters that ultimately form the central SMAC (cSMAC) [5]. The spatially distinct area surrounding the cSMAC is the peripheral SMAC (pSMAC) comprising adhesion proteins that are trans-activated by the TCR signaling and play a major role in maintaining the stability of the IS. The pSMAC also establishes an intimate contact between the two cells, increasing the relative concentrations of secreted molecules in the cleft facilitating the exchange of signals between them. Integrin, the major adhesion protein that segregates to the pSMAC, normally rests in an inactive state and upon activation by chemokines, TCR or selectin changes its conformation to an open active configuration via an inside out signaling mechanism. Active integrin further modulates the TCR via outside in signaling mechanisms [6]. Integrin interacts with ICAM on the target cell to form close adhesions.

Depletion of the actin network occurs at the IS following TCR activation within 2 min of cell–cell adhesion and is indispensable for CG fusion because in cells that do not clear actin at the IS there is no exocytosis [7]. PIP3 was shown to dictate actin polymerization at the immune synapse. PIP3 generated from PIP2 by PI3K recruits Dock2 which in turn activates Rac. The effector protein of active GTP bound Rac is WAVE that plays a role in actin polymerization. Dock2 deficient CTLs display small sized synapses in which the segregation of fluorescently tagged Lifeact into an actin ring is clearly visible [8]. Finally, cortical actin recovery terminates CG fusion [9]. While actin is cleared from the center to form an outermost ring in the distal SMAC (dSMAC), signaling proteins accumulate in a central area called the central SMAC (cSMAC).

Exocytosis of perforin and granzyme B (GzmB) containing cytotoxic granules occur within the secretory domain of the cSMAC that is adjacent to the signaling domain. Exocytosis of CGs is one of the best studied aspects of CTL biology.

The accumulation of TCRs at the IS is important to sustain signaling at the IS [10]. It has been suggested in T helper and naïve T cells that differing TCR strengths induce different modes of immune synapses and that stronger TCR stimuli induce longer lasting or more sustained synapses [11]. CTLs form short term synapses because they can rapidly induce granule induced killing of target cells [12,13,14]. Therefore, we studied how different TCR stimuli influences different parameters of IS formation within this timeframe. For that purpose, we used supported lipid bilayers [15,16,17] with differing anti-CD3 concentrations to induce IS formation and investigated the behavior of molecular markers of the different IS zones under these varying stimuli by total internal reflection fluorescence microscopy (TIRFM). We find that the stimulus strength influences specific stages of the multi-step IS formation differently. Under our experimental conditions where we stimulated CTLs with 5, 10, or 20 µg/mL anti-CD3, we observed an increase in the number and size of TCR-CD3 clusters or size and dwell time of the actin ring with increasing stimuli. However, an optimum concentration of 10 µg/mL anti-CD3 induces the fastest adhesion time to the stimulatory bilayers, maximumTCR-CD3 dwell time, and CG fusion. Our findings show that rather than the amount of signaling molecules it is their dwell time at the immune synapse that ensures maximum effector function in CTLs.

## 2. Results

### 2.1. An Optimum TCR Trigger is Required for Maximal CG Fusion Efficiency

There have been reports demonstrating the direct correlation between the strength of the TCR trigger and the capability of CTLs to lyse target cells [18]. Since exocytosis of CGs is one of the most important consequences of IS formation and effector function of CTLs, we investigated the dependency of this process on the strength of the TCR stimulus. Supported planar bilayers containing biotinylated anti-CD3 and ICAM have been described before to image immune synapses [16,19]. We isolated CTLs from a recently generated Granzyme B-mTFP (GzmB-mTFP) knock-in mouse to endogenously label CGs and allowed them to adhere to such bilayers to form an IS. In order to change the strength of TCR signaling in the CTLs we varied the concentration of anti-CD3 in the bilayers while keeping the concentration of ICAM unchanged. Remarkably, the percentage of adhered cells that secreted GzmB within our imaging time of 15 min was highest in cells that were stimulated with the intermediate concentration of 10 µg/mL anti-CD3. In contrast, bilayers containing 5 and 20 µg/mL anti-CD3 were clearly less efficient in inducing CTLs to secrete GzmB (Figure 1a). An in-depth analysis of the number of fusion events per cell revealed that cells stimulated with 10 µg/mL anti-CD3 also had a significantly higher number of fusion events per cell in comparison to the 5 and 20 µg/mL anti-CD3 stimulation conditions (*** *p* < 0.001) (Figure 1b). As a second independent experiment we performed a CD107 degranulation assay in which CTLs were stimulated with 5, 10, and 20 µg/mL anti-CD3 for 2 h at 37 °C. Interestingly, under these experimental conditions we did not find that the 10 µg/mL is the most efficient stimulus for CTLs among the three different stimuli as increasing the concentration of plate bound anti-CD3 marginally, but insignificantly, increases the amount of surface CD107 as quantified by the median fluorescence intensity of CD107 (Figure 1c).

### 2.2. An Optimum TCR Trigger Strength Enables CTLs to Adhere Significantly Faster to Stimulatory Bilayers in Comparison to Lower or Higher Trigger Strengths

Since we found that CTLs induced with an intermediate strength of TCR stimulus exhibited more CG fusion than when induced with a lower and higher strength we asked whether the observed effect on CG fusion is a consequence of faster cell adhesion to the bilayer. Lifeact, a 17-amino-acid peptide, which stains filamentous actin (F-actin), does not interfere with actin dynamics in vivo and in vitro. When coupled with a fluorophore such as RFP or GFP Lifeact can be used to visualize actin dynamics [20]. We electroporated CTLs with Lifeact-mRFP to label the actin network and analyzed the latency of cell adhesion by calculating the delay between the time when the cells were added to the coverslip (*t* = 0) to when the fluorescence footprint of the cell had an area of ≈3 µm^2^ (Figure 2a). We found that the latency of adhesion was 279.49 ± 4.91 s with the 5 µg/mL stimulus, 45.93 ± 3.68 s with the 10 µg/mL stimulus, and 141.58 ± 7.23 s with the 20 µg/mL stimulus (Figure 2b). Thus, an optimum stimulus enables the CTL to adhere the fastest to the stimulatory bilayers. However, the latency between CTL adhesion and CG secretion is not significantly different between the 5, 10, and 20 µg/mL anti-CD3 concentrations (321.43 ± 24.63 s for 5 µg/mL, 284.69 ± 54.77 s for 10 µg/mL, and for 311.22 ± 28.86 s for 20 µg/mL; Figure 2c). This implies that the time required for CTLs to secrete GzmB to the IS once the cells adhere to the stimulatory bilayers is not dependent on the concentration of anti-CD3 that we used as stimuli. The IS formed by stimulation with 5 µg/mL anti-CD3 was significantly more mobile than when formed by 20 µg/mL anti-CD3 containing bilayers (Appendix A). Immune synapses formed using 5 µg/mL anti-CD3 were more mobile with a velocity of 0.284 ± 0.035 µm·s^−1^ in comparison to what was seen with the 10 µg/mL (0.251 ± 0.027 µm·s^−1^) and 20 µg/mL (0.167 ± 0.018 µm·s^−1^).

### 2.3. While the Dwell Time of Actin Rings Proportionally Increases with the Strength of the TCR Stimulus, the Dwell Time of CD3-mTFP Clusters at the IS Requires an Optimum TCR Stimulus in an Effector CTL Synapse

Since there appears to be an optimum anti-CD3 concentration for inducing the fastest adherence of CTLs to the stimulatory bilayers, we investigated the dependency of other stages of IS formation on an optimum strength of TCR trigger. Actin clearance to form a ring at the dSMAC is an early key step in IS formation initiating first recycling endosome and second cytotoxic granule exocytosis [7,21]. TCR-containing recycling endosomes polarize to the cSMAC upon proximal TCR triggering [10]. We electroporated CTLs with Lifeact-mRFP and CD3-mTFP and were able to visualize actin clearance and cSMAC formation in CTLs upon contact with stimulatory bilayers containing 5, 10, or 20 µg/mL anti-CD3 (Figure 3a, Appendix A).

First, we analyzed the latency of actin ring formation following cell adhesion to the stimulatory bilayers with varying anti-CD3 concentrations and surprisingly found no significant effect. The latency of actin ring formation following adherence was defined as the time interval between cell adherence (*t* = 0) and the first appearance of a stable actin ring. The time of cell adherence was again defined as before (see Figure 2). With the 5 µg/mL condition the latency of actin ring formation was 102.21 ± 5.47 s, for the 10 µg/mL condition it was 105.63 ± 10.04 s, and for the 20 µg/mL condition it was 109.73 ± 8.21 s. No significant differences among the data sets could be demonstrated (Figure 3b). However, CTLs showed a significant increase in the dwell time of the actin ring which was 266.56 ± 11.6 s, 332.36 ± 7.1 s, and 325.3 ± 4.6 s with the 5, 10, and 20 µg/mL anti-CD3 containing bilayers, respectively (Figure 3c). The maximum surface area of the actin ring formed did not significantly change upon increasing anti-CD3 concentrations and was on average 116.97 ± 12.8 µm^2^, 133.13 ± 5.48 µm^2^, and 137.86 ± 10.95 µm^2^ for the 5, 10, and 20 µg/mL anti-CD3 containing bilayers, respectively (Figure 3d). Since varying the strength of the TCR stimulus only appears to influence the dwell time of the actin ring we next analyzed how TCR strength affects specific parameters pertaining to accumulation of TCRs at the IS. The number of CD3-mTFP aggregates per cell were on average 4.27 ± 0.36, 5.83 ± 0.18, and 5.9 ± 0.23 when stimulated by the 5, 10, and 20 µg/mL anti-CD3ᵋ containing bilayers, respectively (Figure 3e). Next, we analyzed CTLs for the average size of a CD3-mTFP aggregate and found that it was 0.84 ± 0.06 µm^2^, 1.48 ± 0.06 µm^2^, and 1.24 ± 0.05 µm^2^ for the 5, 10, and 20 µg/mL anti-CD3 stimulatory bilayers (Figure 3f). Both the size and the number of CD3-mTFP clusters formed in CTLs stimulated with either 5, 10, or 20 µg/mL anti-CD3 were significantly different from one another and increased with increasing stimuli. Finally, we analyzed the dwell time of a CD3-mTFP cluster and found that the cells that adhered to bilayers containing 10 µg/mL anti-CD3 exhibited the longest dwell time at the IS that was significantly different from that observed with 5 and 20 µg/mL anti-CD3. The average dwell time of the CD3-mTFP observed was 99.69 ± 5.3 s, 154.13 ± 7.55 s, and 106.7 ± 4.55 s for the 5, 10, and 20 µg/mL anti-CD3 containing stimulatory bilayers (Figure 3g).

We conclude that the dwell time of the actin ring does not require an optimum TCR trigger but rather increases with the strength of the TCR stimulus. The amount and dwell time of recycling CD3 show different dependencies on the strength of the TCR stimulus. The dependency of the dwell time of CD3-mTFP on the strength of the TCR stimulus correlated with that found for CG exocytosis.

### 2.4. Modes of Exocytosis of Cytotoxic Granules at Immune Synapses Are Independent of the Strength of the TCR Stimulus

Finally, since there appears to be an optimum TCR trigger required for maximal CG fusion efficiency we asked whether this could result in or be a consequence of different modes of exocytosis. In neurons there are reports of fast and slow modes of synaptic vesicle membrane fusion and retrieval depending on the strength of the stimulus, temperature, and synaptic maturation [22]. We could previously demonstrate that Synaptobrevin2 is the major v-SNARE in cytotoxic granules of murine CTLs and that recycling of cytotoxic granule membrane components is essential for multiple killing of target cells [13,23]. Chang et al. speculated there may be differences in the requirement of granule recycling during acute and chronic infections due to differences in the viral load [13]. Therefore, we investigated the temporal dynamics of CG membrane proteins at the IS upon fusion in CTLs stimulated with differing stimulatory anti-CD3 concentrations. CTLs isolated from GzmB-mTFP knock-in mice were electroporated with Synaptobrevin2-pHuji to visualize the lumen and membrane of cytotoxic granules, respectively. Cells were then added to lipid bilayers containing different concentrations of anti-CD3 antibody (5, 10, and 20 µg/mL) and allowed to form an IS. Cells were imaged for 15 min by TIRF microscopy. Both fluorophores were sequentially excited by a 445 nm and a 561 nm laser to detect GzmB-mTFP and Synaptobrevin2-pHuji, respectively. pHuji is a pH-dependent fluorophore that only fluoresces at neutral to basic pH. As the lumen of a CG is acidic, Synaptobrevin2-pHuji is nearly invisible prior to exocytosis but it lights up as soon as the fusion pore of a CG opens. The following diffusion of Synaptobrevin2-pHuji from the CG membrane to the plasma membrane results in a decrease of its fluorescent signal. Additionally, as the fusion pore opens GzmB-mTFP secretion occurs, which is measured as a sudden drop in fluorescence intensity [24]. We observed two different behaviors of Synaptobrevin2-pHuji during the decay phase. In the first type the fluorescence of Synaptobrevin2-pHuji disappeared in less than 1 s (Figure 4a,b, Appendix A). In the second type of behavior Synaptobrevin2-pHuji remained at the same spot where GzmB-mTFP secretion occurred and its fluorescence intensity decreased with a rather long time constant, which was greater than that of short fusion events by a factor of almost 1000 (Figure 4a,b, Appendix A).

We quantified the decay time (tau) of Synaptobrevin2-pHuji fluorescence for the short and long fusion events (see Section 4). For all short fusion events the tau value was on average 0.34 ± 0.05 s, 0.34 ± 0.04 s, and 0.32 ± 0.04 s for the 5, 10, and 20 µg/mL anti-CD3 conditions (Figure 4c). The tau value for the long fusion events was as expected much larger than the short fusion events (379.97 ± 122.47 s, 322.95 ± 134.73 s, and 362.04 ± 172.90 s for the 5, 10, and 20 µg/mL anti-CD3 stimulation conditions, respectively. The tau values of GzmB-mTFP were not significantly changed upon varying stimuli (0.281 ± 0.004 s for the 5, 0.280 ± 0.005 s for the 10, and 0.285 ± 0.005 s for the 20 µg/mL anti-CD3 stimulation; Appendix A).

The complete fusion event duration for the long fusion events was much larger than for the short fusion events (Appendix A). However, there was no significant difference in the time taken to complete the short or long fusion events upon varying anti-CD3 concentrations as specified in the figure (short fusion events: 1.35 ± 0.06 s, 1.46 ± 0.07 s, and 1.48 ± 0.07 s; long fusion events: 536.70 ± 81.64 s, 462.67 ± 122.47 s, and 570.60 ± 127.60 s). Out of the 31 fusion events analyzed for the 5 µg/mL anti-CD3 conditions, 80.6% were of the short type and 19.4% were of the long type. For the 10 µg/mL anti-CD3 stimulation 82.4% of the 34 fusion events were short fusion events and 17.6% were long fusion events. Lastly, for the 20 µg/mL anti-CD3 condition 88.2% of the 34 fusion events were short fusion events and 11.8% were long fusion events (Figure 4d). The fusion pore opening times for both fusion event types are similar as shown in Figure 4e and unaffected by varying the strength of the TCR trigger (0.55 ± 0.07 s, 0.66 ± 0.03 s, and 0.65 ± 0.03 s for 5, 10, and 20 µg/mL anti-CD3 stimulation; Figure 4e).

Therefore, our results show that we can distinguish two distinct modes of fusion events which do not differ in the decay time of GzmB-mTFP but show a large difference by a factor of 1000 in the decay time of Synaptobrevin2-pHuji. The kinetics of these two types of fusion events appear to be independent of varying strengths of TCR trigger.

## 3. Discussion

We show here that the different steps of IS formation are differently dependent on the strength of the stimulus. We generated effector CTLs by stimulating them with anti-CD3/anti-CD28 coated beads and IL-2 to study the consequences of varying strengths of TCR stimuli of cytotoxic synapses on IS formation and effector function. We used supported planar bilayers with varying concentration of biotinylated anti-CD3 but constant ICAM concentrations as surrogate targets to trigger granule release in CTLs. Immune synapses formed between CTLs and the stimulatory bilayers were imaged at a high temporal and axial resolution by TIRFM. We found that increasing the strength of the TCR trigger does not automatically result in a linear increase in downstream stages of IS formation and that certain stages of IS formation are more sensitive to the initial TCR trigger than others. Under our experimental conditions we found that with an optimum TCR trigger consisting of 10 µg/mL anti-CD3 in the stimulatory bilayers, CTLs exhibit the shortest latency time for adhesion to stimulatory bilayers, the longest dwell time of CD3-TFP clusters and maximal CG fusion.

We used anti-CD3 as a stimulus in our study which is not as physiological as a cognate peptide-MHC complex stimulation. Biotinylated anti-CD3 on supported planar bilayers have been used to reliably report downstream signaling events in T cells with clear segregation of the IS into SMACs comparable to that observed with a peptide MHC stimulation [19]. We also have evidence that CTLs stimulated with anti-CD3 vs. cognate peptide-MHC complexes behave similarly in our functional readouts (see materials and methods in [13]. Physical parameters such as mechanical forces, however, do influence agonist TCR-pMHC interactions occurring during T cell priming processes [25]. While this is something we cannot address in our study, it would provide more information about the function of the cytolytic IS.

Our study describes the influence of the initial TCR trigger on the formation and efficiency of a cytotoxic IS. We cannot exclude that under physiological conditions in vivo varying stiffness of target cell membranes could also serve as a variable that affects IS dynamics. Basu et al. cultured B16 melanoma cells on stiff and soft hydrogels to induce a higher and lower membrane tension and could demonstrate a clear increase in sensitivity to perforin mediated lysis on the target cell membranes with higher tension [26]. CD4^+^ T cells also exhibit increased sensitivity to varying stiffness of activation substrates. Upon activation for 24 h with biotinylated anti-CD3, anti-CD28, and ICAM coated poly-acrylamide (PA) gels with increasing stiffness values of 0.5, 6.4, and 100 kPa, maximum activation induced surface marker expression and cytokine production was seen in CD4^+^ T cells that were stimulated with the stiffest activation substrates [27].

Many reports have demonstrated that increasing concentrations of cognate antigen increase CTL mediated cytotoxicity and cytokine release [18]. Our TIRFM data demonstrate that under an optimum TCR stimulus, CTLs exhibit maximal cytotoxic granule fusion. Increased CTL mediated lysis upon increasing antigen strength may also be a consequence of alternate pathways being induced such as FAS mediated killing pathways. Interestingly, an optimum stimulus does not seem to be required for lysosomal degranulation even though surface LAMP1 detection has been widely used as a readout for CG exocytosis [28]. One reason for this observation may be that LAMP1 degranulation assays are also detecting lysosomal degranulation occurring over the entire cell surface and not only at the IS. Another reason may be that in a LAMP1 degranulation assay the cells are stimulated for more than 2 h at 37 °C thereby giving CTLs enough time to adhere to the coated anti-CD3 surface and exhibit CG exocytosis. Xhao et al. showed that CTLs exhibit a bell-shaped calcium dependency for perforin mediated killing [29]. According to them an intracellular Ca^2+^ concentration or [Ca^2+^]_i_ between 122 and 334 nM and an extracellular Ca^2+^ concentration or [Ca^2+^]_o_ between 23 and 625 µM is ideal for CTL and NK killing related effector functions. Although intracellular calcium levels under our varied TCR stimuli have not been tested an increase in calcium influx upon higher antigen stimulation has been demonstrated [30]. Since our results demonstrate that there is a significant decrease in CG fusion beyond an optimum anti-CD3 concentration (in our case 10 to 20 µg/mL) our observations may be explained by the bell-shaped Ca^2+^ dependence.

Different granule fusion modes essentially control the kinetics and amount of the cargo released [31]. Various modes of lytic granule fusion have been observed in NK cells and CTLs differing in the amount of cargo release or diffusion of membrane bound LAMP1 [32,33]. Our results with CG membrane bound Synaptobrevin2-pHuji and CG cargo Granzyme B-mTFP demonstrate 1000-fold differences in the decay times of Synaptobrevin2-pHuji fluorescence at the site where granzyme B is released entirely. The different decay times may be due to different rates of retrieval of Synaptobrevin2 into acidic compartments. Unexpectedly, we did not observe any dependency of these decay times on the various concentrations of stimulatory anti-CD3 in our experiments. We might be able to gain more insight into the various fusion modes and their relevance by investigating CTLs generated from differing strengths or exposure time of stimuli on naïve CD8^+^ T cells in the future.

Increasing TCR strength has a positive effect on the size of CD3 clusters accumulating at the IS. Upon a TCR trigger, TCR clusters initially form at the pSMAC and move towards the cSMAC. These clusters then coalesce to form bigger clusters [19]. Our results demonstrate that with an increasing stimulus there is an increase in the size and number of average CD3 clusters. However, there is an optimal concentration of anti-CD3 that elicits a significantly longer dwell time of the CD3-mTFP at the IS. Therefore, the size and dwell time of CD3 clusters are differently dependent on the strength of the TCR stimulus. It is well known that sustained TCR signaling is required for synapse maintenance and full effector function of naïve helper T cells [34]. Our observation of a significant increase in the dwell time of CD3-mTFP at the IS in CTLs stimulated with an optimum 10 µg/mL anti-CD3 may explain the maximum fusion efficiency exhibited by these CTLs.

The varied stimulatory anti-CD3 concentrations in the bilayers do not influence the latency of actin ring formation in CTLs following adhesion. Since central clearance of actin plays a key role in controlling the timing of CG release at the synapse [7], this could explain our results in which we do not find a significant difference in the latency of GzmB secretion following adhesion of the CTL to the bilayers with varied anti-CD3 concentrations. Le Floc’h et al. reported that Dock2-deficient CTLs display miniaturized synapses [8]. Since our results show that the overall surface area of the actin ring formed is not altered upon changing TCR trigger strengths we conclude that PIP3 recruited, Dock2-dependent actin polymerization was not affected. This explains why we also do not find a significant difference in the latency of actin ring formation following adhesion upon changing anti-CD3 concentrations. However, the dwell time of the actin ring is increasing upon increasing TCR strength implying that increasing the TCR trigger is inducing more stable actin rings and in turn more stable synapses. It has been demonstrated that LAT is not required for kinetics of IS formation but for maintaining stable synapses in CTLs [35]. Since our results demonstrate that the dwell time of the actin ring and CD3-mTFP show different dependencies on the stimulatory anti-CD3 concentrations this may be due to LAT-induced downstream effects.

T cells adhering to stimulatory bilayers indeed tended to display more movement with lower anti-CD3 concentrations. Since increasing mobility reduces formation of a stable adhesion it may explain why cells stimulated with 5 µg/mL anti-CD3 have lower fusion efficiency and fusion events per cell. However, the cells stimulated with 20 µg/mL anti-CD3, although showing less movement, also demonstrate reduced fusion efficiency. Thus, the formation of a stable adhesion with reduced movement is not a cause but rather a consequence of high antigen trigger that may lead to inefficient T cell effector function probably due to exhaustion. CTL exhaustion has been described and occurs in cases of prolonged antigen exposure such as in chronic infections and cancers during T cell activation [36]. Under our experimental settings we are imaging CTL synapses induced by different anti-CD3 concentrations for 10 min. It would be rather surprising if we are inducing CTL exhaustion just upon higher antigenicity and it might provide new mechanistic insights into the exhaustion process itself. Since T cell exhaustion occurs prominently in cancers, understanding the mechanism of exhaustion may help us understand how to reverse it which will open new doors to cancer immunotherapy [37].

## 4. Materials and Methods

### 4.1. Mice

Granzyme B-mTFP knock-in mice were generated with the CRISPR CAS9-D10A_mRNA (Nickase) system in cooperation with the Max-Planck-Institute for experimental Medicine in Göttingen. As a guide RNA the GzmB-protospacer-sgRNA2 sequence 5′ GTC CAG GAT TGC TCT AGG AC 3′ was used and a Homology-Directed-Repair fragment of GzmB-mTFP was inserted. Afterwards the genome of the mouse was fully sequenced to exclude side effects. All wild type mice (obtained from Charles River Laboratories, Sulzfeld, Germany) used for experiments were between 8 and 12 weeks old and belonged to the C57Bl/6N (Black 6) strain. All animal experiments were approved by the state of Saarland (Landesamt für Gesundheit und Verbraucherschutz; animal license number 41-2016; approval date: 8 November 2016) and were in accordance with German law and European animal healthcare guidelines (FELASA).

### 4.2. Cells

Naïve CD8^+^ T cells were isolated from splenocytes using a FlowComp Dynabeads Mouse CD8^+^ positive isolation kit (ThermoFisher Scientific, Karlsruhe, Germany) as described previously [13]. The cells were cultured at a concentration of 1 × 10^6^ cells per mL in AIMV Medium (ThermoFisher Scientific) supplemented with 10% FCS, 0.5% Pen/Strep and 40 µM BME (Carl Roth, Karlsruhe, Germany). Then, 2 × 10^6^ cells were seeded per well of a 24-well plate and the cells were activated with anti-CD3/anti-CD28 mouse activator beads at a ratio of 1:0.8. Cells were incubated at 37 °C with 5% CO_2_ for 7 or 8 days to generate effector CD8^+^ T cells or CTLs. Cells were counted every day after day 2 of activation and split to 1 million/mL as the cells doubled or tripled. Furthermore, 10 U/mL of recombinant mouse IL2 (BD Biosciences, San Jose, CA, USA) was added as supplement to each well during the splitting. Cells activated for 7 or 8 days as described above were used for all experiments.

### 4.3. Electroporation of CTLs

Electroporation was performed using the Nucleofection kit (LONZA, Cologne, Germany) according to the manufacturers protocol. Briefly, for one electroporation 5 × 10^6^ CTLs from WT or granzyme B-mTFP knock-in mice were centrifuged at 180× *g* for 7 min, resuspended in 4 mL of isolation buffer and spun again at 180× *g* for 7 min. The cell pellet was finally resuspended in 100 µL nucleofection solution and 1.5 µg of each corresponding plasmid DNA was added before subjecting the cells to electroporation using program X-001. Following electroporation, the cells were kept in medium provided by LONZA overnight at 32 °C with 5% CO_2_ in one well of a 12-well plate. After 12 h of incubation the cells were spun again for 7 min at 180× *g* and then resuspended in 3 mL of AIM V medium with 10 U/mL recombinant mouse IL2 and again placed in the incubator at 32 °C with 5% CO_2_ until the end of imaging.

### 4.4. Plasmids

The Lifeact-mRFP construct was kindly provided by Roland Wedlich-Söldner (Universität Münster, Münster, Germany). CD3-mTFP was described previously [38]. pHuji was amplified from a previously described VAMP7 pHuji construct [39] using the forward primer 5′ ATG TAT ACC CAA GCT TAT GGT GAG CAA GGG CGA G 3′ to add HindIII site and the reverse primer 3′ ATG TAT ACG CGG ATC CTT ACT TGT ACA GCT CGT C 5′ to add the BamHI site. The PCR product was digested and ligated into an optimized pMax-vector containing Synaptobrevin2 as previously described [40].

### 4.5. Lipid Bilayer Preparation

Preparation of proteins: first, 0.05 μg/mL biotin (EZ-link sulfo-NHS-LC-LC-Biotin (ThermoFisher Scientific)) was added to an anti-CD3 antibody solution (clone 145-2C11 diluted to 1 mg/mL) and incubated for 30 min at room temperature. Overnight dialysis at 4 °C in PBS using a dialysis cassette (Slide-A-Lyzer^®^ Dialysis Cassettes, ThermoFisher Scientific) was performed for purification. Mouse ICAM-1 was generated from a drosophila S2 cell line expressing ICAM-1 (kindly provided by Michael Dustin, University of Oxford, UK). The protein was purified after S2 induction. Preparation of lipids: first, 100% DOPC (#850375C Sigma Aldrich/Avanti (Taufkirchen, Germany), 25% NTA (#790404C Sigma Aldrich/Avanti) and 2% capbio (#870273C Sigma Aldrich/Avanti) were prepared in a lipid buffer containing Tris (25 mM, pH 8) and NaCl (150 mM) bubbled with N_2_ as the working stock solution. All lipids were dissolved in DMSO (Avanti). The lipids were then dried in a lyophilizer for 2 h at −40 °C and finally resuspended in the lipid buffer to yield a liposome suspension with inhomogeneous sizes of vesicles. The suspension was extruded in avestin extruder (Lipofast^®^) to obtain homogenously sized SUVs (single unilamellar vesicles). The lipids were diluted to a concentration of 0.4 mM each with DOPC. The glass slides were pre-cleaned with piranha solution (50 mL of concentrated sulfuric acid + 25 mL of hydrogen peroxide), plasma cleaned to remove organic contaminations, and placed onto reaction chambers (Ibidi stickyslide^®^ VI microscopy chamber). Coating of lipids and proteins on glass slides: the liposome solution was applied to the reaction chamber. Then, 100 μM NiSO_4_ in 5% casein was applied followed by ICAM His-tag (0.275 μg/mL). In the final step the biotinylated anti-CD3 antibody was applied depending on the concentration needed (5, 10, or 20 μg/mL) for the experiment. The concentration of capbio lipids was also varied according to the concentration of the anti-CD3 antibody used. Washing of wells after each step was done with HBS/HSA buffer.

### 4.6. Total Internal Reflection Fluorescence Microscopy

The TIRFM setup used was described previously [39] with the following changes: a 445 nm laser (100 mW) along with a 561 nm solid state laser (100 mW) were used for excitation along with a filter cube consisting of an emission filter ZET 442/514/568 and a dichroic beamsplitter ZT405/440/514/561 (Chroma Technology Corp, Olching, Germany). The pixel size of the camera was 160 nm. The acquisition frequency for all experiments was 5 Hz and the acquisition time was 15 min. In TIRF mode the penetration depth of the evanescent light for the 561 nm and the 445 nm laser was set at 150 nm through the iLAS2 illumination control system (Roper Scientific SAS, Evry, France) for all experiments. Lipid bilayers were always freshly prepared on the day of the imaging experiments. Before the addition of cells and imaging, the HBS/HSA buffer was removed from the well of the prepared reaction chamber and 200,000 CTLs initially suspended in a low calcium imaging buffer (155 mM NaCl, 4.5 mM KCl, 5 mM HEPES, 3 mM MgCl2 × 6 H_2_O, and 10 mM glucose) were added to the bilayers and imaged for 5 min. Cells were then perfused with a high calcium imaging buffer (140 mM NaCl, 4.5 mM KCl, 5 mM HEPES, 2 mM MgCl2 × 6 H_2_O, 10 mM CaCl_2_, and 10 mM glucose) to induce exocytosis and then imaged again for 10 min.

### 4.7. Imaging Analysis

All image analysis was done using ImageJ 1.52n or IGOR PRO (WaveMetrics, Portland, OR, USA). For analysis of the latency of accumulation a stable adhesion was defined as the fluorescent signal having an area of ≈3 µm^2^. The analysis of latency time of the first formed actin ring, its dwell time, and maximum area were analyzed manually after setting thresholds to images to distinguish fluorescence from noise and sometimes verified by the line plot function. The area of the actin ring was determined for all time frames and the maximum among all time frames was calculated for each concentration to analyze the area of cell spread at the IS. The CD3-mTFP cluster size and number and dwell time of vesicles was also analyzed automatically using the analyze set measurement parameters. Dwell time is defined as the time during which the fluorescent CD3-mTFP vesicle or the actin ring remains in the TIRF field. To determine the mobility of the IS, i.e. the center of mass of the cell footprint, the IS was determined every 200 ms for the entire recording. The change in the x and y coordinates could be reshaped into lines that form the two sides of a right triangle. The hypotenuse could be calculated using the Pythagoras theorem to generate the mobility of the IS every 200 ms and subsequently every second. All the analysis concerning the kinetics of the short and long fusion events were done using IGOR PRO. The decay time of Synaptobrevin2-pHuji fluorescence was quantified by fitting the curve with a single exponential and then calculating the time taken for 67% of decrease in fluorescence to obtain the tau (τ). All results were split into two groups based on the tau values. All fusion events with a tau value of less than a second (τ < 1 s) were categorized as short fusion events and all fusion events with a tau value of more than a second (τ > 1 s) were categorized as long fusion events. The complete fusion event time for both short and long fusion events was quantified by calculating the time elapsed between the beginning and the end of the entire fusion event. The fusion pore opening time for all fusion event types was quantified by fitting with a line the sharp increase phase of Synaptobrevin2-pHuji fluorescence and the 10% to 90% rise time was calculated.

### 4.8. Degranulation Assay

Degranulation assay was basically as described previously [40] with the following changes. Briefly, day 7 or 8 activated CTLs were rested for 2 h without any stimulus in fresh culture medium. Afterwards, 0.5 million cells were resuspended in 100 μL culture medium with 1 µg/mL anti-CD107 PE (clone 1D4B, Biolegend, San Diego, CA, USA) per well of a 96-well plate. The cells were plated onto wells pre-coated with either 5, 10, or 20 μg/mL anti-CD3^ε^ (clone 145-2C11, BD Biosciences) or not (constitutive controls) and incubated for 2 h at 37 °C with 5% CO_2_. Unstained controls were performed for every concentration of anti-CD3 used as stimulus to ensure no major change in the auto-fluorescence of the cells upon stimulation with varying anti-CD3 concentrations. Cells were then washed twice with ice cold PBS and analyzed by BD FACSAria^TM^ III. Data were analyzed by using FlowJo software (Celeza, Switzerland). Median fluorescent intensity (MFI) of CD107-PE was calculated for the constitutive and stimulated conditions and degranulation was analyzed as the fold change in MFI of CD107.PE upon stimulation normalized to constitutive levels.

### 4.9. Statistics

One-way ANOVA test was used to test significance for data that was normally distributed and ANOVA on Ranks with the post hoc Dunn’s method for data that was not normally distributed. All statistical tests were done using SigmaPlot version 13.0 (Systat Software, Erkrath, Germany).

## Figures and Tables

**Figure 1 ijms-21-02475-f001:**
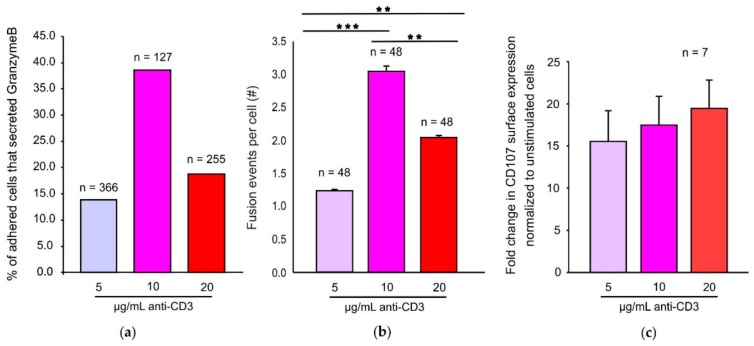
An optimum TCR trigger is required for maximal cytotoxic granule (CG) fusion efficiency in supported planar bilayers. (**a**) Percentage of adhered cells that secreted granzyme B. *n* = number of adhered cells. (**b**) Number of fusion events per cell when stimulated with 5, 10, or 20 µg/mL anti-CD3. *n* = number of cells that secreted granzyme B. (**c**) Fold change in CD107 degranulation normalized to constitutive degranulation occurring upon varying stimuli. *n* = number of experimental repeats. Error bars represent mean ± SEM. ** *p* < 0.01; *** *p* < 0.001; ns, non-significant (*p* > 0.05).

**Figure 2 ijms-21-02475-f002:**
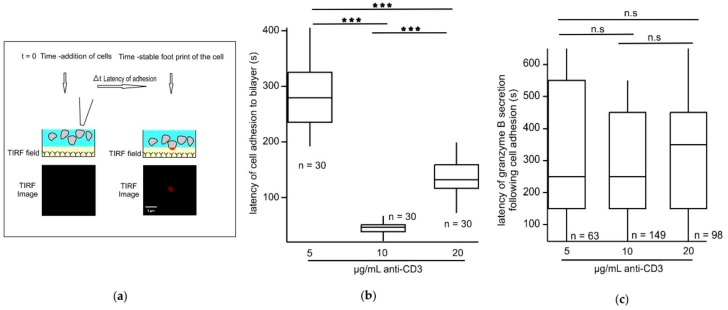
An optimum strength of TCR trigger enables cytotoxic T lymphocytes (CTLs) to adhere significantly faster to stimulatory bilayers. (**a**) Cartoon depicting the experimental strategy to measure latency of adhesion. (**b**) Box plots depicting the latency of cell adhesion to stimulatory bilayers with varied anti-CD3 concentrations. *n* = number of cells analyzed. (**c**) Box plots depicting the latency of granzyme B secretion normalized to the time of cell adhesion to stimulatory bilayers with varying anti-CD3 concentrations as mentioned in the figure. *n* = number of vesicles analyzed. *** *p* < 0.001; ns, non-significant (*p* > 0.05).

**Figure 3 ijms-21-02475-f003:**
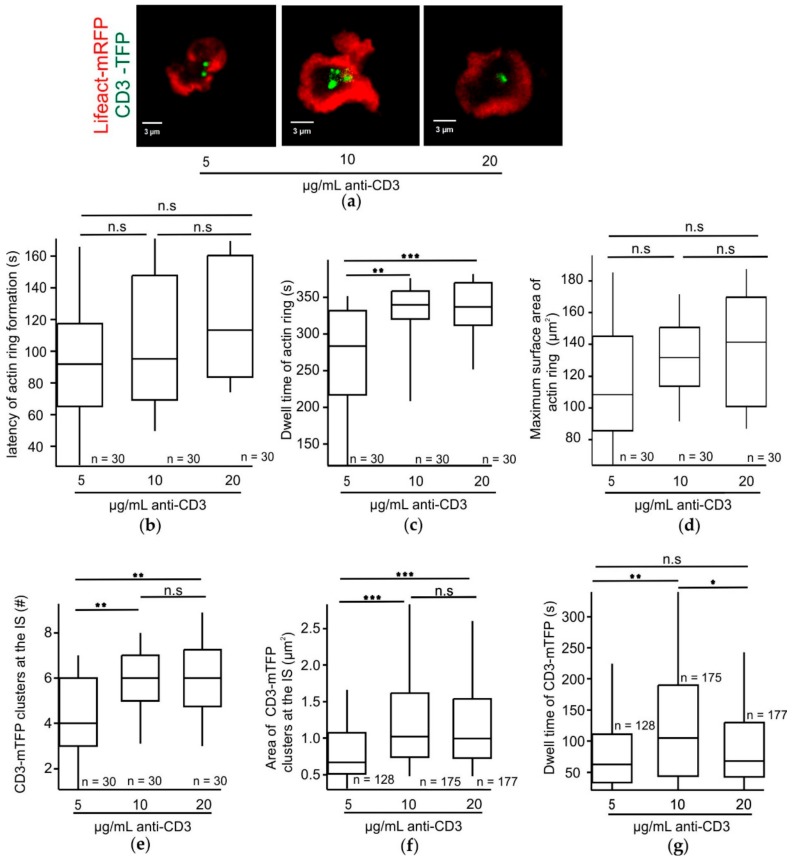
The dwell time of actin rings proportionally increases with the strength of the TCR stimulus, but the dwell time of CD3-mTFP clusters at the immune synapse (IS) requires an optimum TCR stimulus in an effector CTL synapse. (**a**) CTLs expressing Lifeact-mRFP (red) and CD3-mTFP (green) adhered to form an IS with varying anti-CD3 concentrations as trigger. Box plots depicting (**b**) latency of actin ring formation, (**c**) dwell time of actin ring, and (**d**) maximum surface area of dSMAC formed. *n* = number of cells analyzed for each concentration of anti-CD3 stimulus. Box plots depicting (**e**) number of CD3-mTFP clusters where *n* = number of cells analyzed, (**f**) average area, and (**g**) dwell time of CD3-mTFP clusters. *n* = number of CD3 clusters analyzed. * *p* < 0.05; ** *p* < 0.01; *** *p* < 0.001; ns, non-significant (*p* > 0.05).

**Figure 4 ijms-21-02475-f004:**
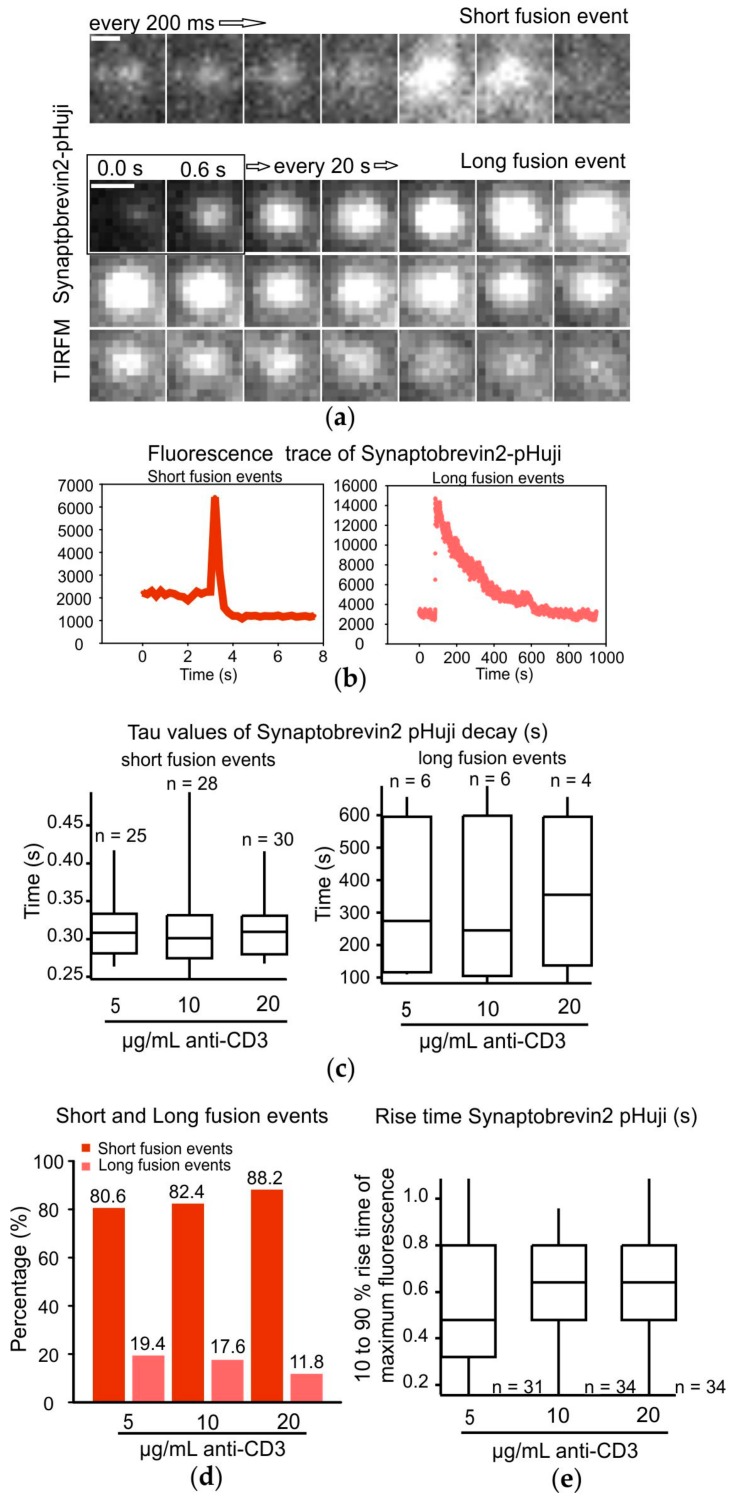
Modes of exocytosis of cytotoxic granules at immune synapses are independent of the strength of the TCR stimulus. (**a**) Exemplary total internal reflection fluorescence microscopy (TIRFM) image snapshots every 0.2 or 20 s unless specified otherwise to visualize Synaptobrevin2-pHuji during short and long cytotoxic granule fusion events, respectively. Scale bars: 1 µm. (**b**) Exemplary fluorescence intensity time trace analysis of typical short and long fusion events using Synaptobrevin2-pHuji as the fluorescent marker of cytotoxic granules. (**c**) Decay time of Synaptobrevin2-pHuji in the short (left) and long (right) fusion events in CTLs stimulated with 5, 10, or 20 µg/mL anti-CD3. *n* = number of fusion events. (**d**) Percentage distribution of short and long fusion events occurring in CTLs when stimulated with 5, 10, or 20 µg/mL anti-CD3 to induce effector immune synapses. (**e**) Rise time of Synaptobrevin2-pHuji in the short and long fusion events in CTLs stimulated with 5, 10, or 20 µg/mL anti-CD3.

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
