# Peer review of "Various Stages of Immune Synapse Formation Are Differently Dependent on the Strength of the TCR Stimulus"

_ijms, 2020, doi:10.3390/ijms21072475_

Round 1
Reviewer 1 Report
Reviewer`s comment;
In this article, Dr. Estl and his colleagues described how different concentrations of antigen stimulus change the formation of IS structure. They defined the optimum concentration for TCR stimulation. There are a few points that require revisions.
Major revision;
- In this article, they stimulated T cells with different concentrations of anti-CD3 on the lipid bilayer. However, anti-CD3 stimulation is an artificial way to activate T cells and is not sufficient to mimic the physiological conditions of the immune system. To overcome this problem, peptide-bound MHC may be added into the lipid bilayer to activate T cells.
Minor revisions;
- The abstract is written improperly. For example, they did not use abbreviations for CTL and CG. Abbreviations may be written as `immune synapse (IS)`. More importantly, the abstract part is not informative enough to explain the article briefly. Please re-write and use clear explanations.
- At line 229, the Tau values of synaptobrevin2-pHuji are labeled as Figure 4e. However, it is presented in panel c. Please check it.
- In section 2.4, they mentioned about the short and long fusion event. However, they did not explain why they checked these events. Please define their biological roles and their importance for CTLs.
- In the discussion section, they mentioned LAT and its possible effect on the stability of IS formation. It would be a great support to this article if they provided any LAT-related data. Although the paper is well-written, it is still straightforward and requires more details.
Author Response
In this article, Dr. Estl and his colleagues described how different concentrations of antigen stimulus change the formation of IS structure. They defined the optimum concentration for TCR stimulation. There are a few points that require revisions.
Major revision:
In this article, they stimulated T cells with different concentrations of anti-CD3 on the lipid bilayer. However, anti-CD3 stimulation is an artificial way to activate T cells and is not sufficient to mimic the physiological conditions of the immune system. To overcome this problem, peptide-bound MHC may be added into the lipid bilayer to activate T cells.
The reviewer is correct that anti-CD3 stimulation does not represent a physiological stimulus. However, we have in the past compared efficacy of CTLs in a redirected lysis assay using anti-CD3 to crosslink them with P815 target cells vs in a more specific cognate peptide MHC stimulation assay. We obtained qualitatively similar results under our readout parameters (see materials and methods in Chang et al., 2016). Since we have only 10 days to submit the revision, new experiments are not possible. To address the reviewers concern we have, however, discussed this issue (lines 291-299).
Minor revisions:
- The abstract is written improperly. For example, they did not use abbreviations for CTL and CG. Abbreviations may be written as `immune synapse (IS)`. More importantly, the abstract part is not informative enough to explain the article briefly. Please re-write and use clear explanations.
We have rewritten the abstract and tried to explain more clearly the basis and the findings of the article. In addition, we have now explained abbreviations clearly as requested.
At line 229, the Tau values of synaptobrevin2-pHuji are labeled as Figure 4e. However, it is presented in panel c. Please check it.
We have corrected the mislabeling (line 255).
In section 2.4, they mentioned about the short and long fusion event. However, they did not explain why they checked these events. Please define their biological roles and their importance for CTLs.
The reviewer is right in that we have not mentioned the reason for checking the long and short events in CTLs. We have included the reasoning behind checking the dynamics of Synaptobrevin2-pHuji decay in lines 222-225 in the section. Additionally, we have included a section in the Discussion (lines 330-340) discussing this topic with some relevant literature.
In the discussion section, they mentioned LAT and its possible effect on the stability of IS formation. It would be a great support to this article if they provided any LAT-related data. Although the paper is well-written, it is still straightforward and requires more details.
The reviewer is right in that LAT-related data might greatly support our article. However, at this point we do not have it and we will also not be able to provide it within the next 10 days. However, ongoing work on this topic will be done with LAT probably leading to mechanistic insights.
Reviewer 2 Report
The manuscript by Estl et al. investigated the variables affecting the immunological synapse (IS) formation between target cell and cytotoxic T cell (CTL) using a aCD3- and ICAM-1-functionalized planar bilayer stimulating mouse T cells which express GzmB-mTFP for tracking CGs. Based on the TIRFM imaging data, they found that the maximum CG fusion was observed only at an optimal TCR signaling strength as established by the concentration of aCD3 (10 μg/ml), although the modes of exo- and endocytosis of CGs is independent of the strength of TCR stimulus. This is a focused, interesting work with a well advanced experimental design, clear data presentation and coherent writing. The conclusion is in general supported by the data presented. Their findings may be of interest to scientists working on the biochemical and biophysical aspects of CTL activation. I only have one comment.
1.Discussion. It cannot be excluded that the target cell stiffness could serve as a variable that affects the dynamics of IS formation. The rigid antigen presentation system used in this study can be 1000 times stiffer than the real cell plasma membrane. It would be helpful if the authors could discuss this aspect in the Discussion section after searching relevant information in the literature.
Author Response
The manuscript by Estl et al. investigated the variables affecting the immunological synapse (IS) formation between target cell and cytotoxic T cell (CTL) using a aCD3- and ICAM-1-functionalized planar bilayer stimulating mouse T cells which express GzmB-mTFP for tracking CGs. Based on the TIRFM imaging data, they found that the maximum CG fusion was observed only at an optimal TCR signaling strength as established by the concentration of aCD3 (10 μg/ml), although the modes of exo- and endocytosis of CGs is independent of the strength of TCR stimulus. This is a focused, interesting work with a well advanced experimental design, clear data presentation and coherent writing. The conclusion is in general supported by the data presented. Their findings may be of interest to scientists working on the biochemical and biophysical aspects of CTL activation. I only have one comment.
1.Discussion. It cannot be excluded that the target cell stiffness could serve as a variable that affects the dynamics of IS formation. The rigid antigen presentation system used in this study can be 1000 times stiffer than the real cell plasma membrane. It would be helpful if the authors could discuss this aspect in the Discussion section after searching relevant information in the literature.
The reviewer is right about the caveats of the rigid antigen presentation system used in our study. We have accordingly included a relevant section in the Discussion (lines 300-310).
Round 2
Reviewer 1 Report
In the revised version of this article, the authors edited the points that I mentioned previously. Introduction and discussion parts are informative enough to express the idea and findings of the article to the readers. Although they did not manage to perform major revision that I mentioned, which is the addition of pMHC into the bilayer system, the explanations in the discussion part are sufficient for this study.